# The Effect of Heat Treatment on Phase Structure and Mechanical and Corrosion Resistance Properties of High Tungsten Ni-W Alloy Coating

Yingjun Xu *, Deyong Wang, Minqi Sheng ⬤, Huihua Wang, Ruiqi Guo, Tianpeng Qu and Shaoyan Hu ⬤

School of Iron and Steel, Soochow University, Suzhou 215000, China; dywang@suda.edu.cn (D.W.); shengminqi@suda.edu.cn (M.S.); hhwang@suda.edu.cn (H.W.); rqguo@suda.edu.cn (R.G.); qutianpeng@suda.edu.cn (T.Q.); syhu616@suda.edu.cn (S.H.)
* Correspondence: xuyingjun0000@gmail.com

**Abstract:** The present study investigated the surface morphology, phase composition, mechanical properties, and corrosion resistance of Ni-W alloy coatings prepared under current densities of 1–5 A/dm$^2$, after undergoing heat treatment at 400 °C, 600 °C, and 900 °C. The grain size of the as-plated Ni-W alloy coating was below 10 nm. After heat treatment at different temperatures, the grain size increased, reaching a maximum value of around 30 nm at 900 °C. Heat treatment crystallized and altered the structure of the coating. Different heat treatment temperatures yielded different precipitates, including Ni$_4$W, Ni$_6$W$_6$C, and WC. The highest coating hardness (820–940 Hv) was achieved at 400 °C, while the best corrosion resistance was achieved at 600 °C. The precipitation hardening phase can be obtained by proper heat treatment temperature, yielding the desired properties of the composite coating.

**Keywords:** nanocrystalline alloys; electrodeposition; high tungsten content; grain growth





## 1. Introduction

Cemented carbide is a high-hardness, wear-resistant composite material made from tungsten carbide particles bonded with a metal binder such as cobalt or nickel, primarily used in cutting tools and wear-resistant components. Its main component is refractory metal carbide (WC, TiC) micron powder with high mechanical strength, with molybdenum (Mo), nickel (Ni), and cobalt (Co) as binders [1]. Cemented carbide has excellent properties, including high hardness, good strength and toughness, wear resistance, corrosion resistance, and heat resistance. As a result, it is widely used in the military industry, aerospace, mechanical processing, metallurgy, petroleum drilling, mining tools, electronics communications, construction, and other areas. Demand for cemented carbide is increasing due to the development of downstream industry [2]. Recently, some researchers have focused on Ni-W alloy coating since its wear resistance, hardness, and corrosion resistance are similar to cemented carbide [3,4].

Heat treatment could improve the hardness and abrasion resistance of Ni-W alloy coating, which could improve the performance of Ni-W alloy coating [5]. Vamsi et al. [6] found that heat treatment affects the microstructure and mechanical properties of pulse electrodeposited Ni-W alloy coatings by inducing amorphous phase crystallization and second-phase precipitation. Additionally, they found that heat treatment improves the coating hardness due to the diffusion reinforcement of precipitation and the barrier effect of grain boundary on dislocation. Some studies have also found that carbon impurities are common contaminants in Ni-W alloy preparation and that carbon plays a key role in the thermal stability of nanocrystalline Ni-W alloy [7–9]. Therefore, the common Ni-W alloy coating was Ni-W-C ternary alloy. The carbide could be precipitated from the coating by heating the Ni-W alloy under certain parameters. WC has been synthesized in situ by laser

cladding, improving the properties of Ni coating and Co alloy coating [10,11]. Su et al. [12] found that the high-frequency induction heat treatment of Ni-W alloy coating yielded $Ni_4W$ and $Ni_6W_6C$ precipitates, achieving a 1100 Hv coat hardness. Lee et al. found that $Ni_4W$ and $Ni_6W_6C$ were obtained by irradiating Ni-W alloy at room temperature and annealing at 850 °C [13].

Hard particles, such as WC [14], TiN [15], TaC [16], SiC [17], $Al_2O_3$ [18], and $ZrO_2$ [19], were added to the Ni-W electrolyte to prepare composite coatings, improving the mechanical properties and corrosion resistance of Ni-W alloy. However, composite plating has several drawbacks, such as high equipment requirements and complex operations. Additionally, increasing the quantity of composite particles in the coating is also challenging. In a study by Yuan [20] et al., WC grains were synthesized in the W-Fe-Ni-C alloy system by an in situ metallurgical reaction. The mechanism underlying the growth of the WC grains was analyzed. Studies have predicted that, to a certain extent, when the appropriate process parameters are used in the Ni-W-C alloy system, the second phase particles, such as $Ni_6W_6C$ and WC, precipitate from the coating and improve the hardness and wear resistance of the composite coating.

Currently, the influence of carbon element in Ni-W alloy coatings is often overlooked, and the impact of carbon element on the heat treatment of Ni-W alloy coatings is even less addressed. Therefore, this study aims to investigate the phase analysis and mechanical corrosion resistance of Ni-W alloy coatings prepared at various current densities under different heat treatment temperatures, with a particular emphasis on examining the role of carbon element in the Ni-W-C ternary system.

## 2. Materials and Methods

The electrodeposition experiment was performed using the DC supplied by a transformer rectifier (4–8 V), where the stainless steel was the anode, while the Q235 steel plate with the size of 200 mm × 10 mm × 2 mm was the cathode. Before the experiment, the steel plate was unoiled using an organic alkaline solution, and then the surface was activated using a 10 wt.% $H_2SO_4$ solution. As shown in Table 1, the Ni-W alloy coatings were prepared by applying pulse current with different pulse parameters at 65 °C from aqueous electrolytes. Next, the plated samples were washed and dried with deionized water. For better phase analysis, the coating was stripped from the substrate to obtain the coating powder sample, eliminating the influence of interdiffusion between the coating and the substrate during heat treatment. The deposited sample and coating powder samples were placed in a vacuum tube furnace and protected by Ar gas. The temperature was increased to 400 °C, 600 °C, and 900 °C at 10 °C/min and retained for 2 h. The specimens were furnace cooled to room temperature and then removed for testing. The composition of the bath and operating conditions are shown in Table 1.

**Table 1.** Ni-W alloy electrolyte and electroplating parameters.

| Electrolyte | Content (g/L) | Function |
| --- | --- | --- |
| $NiSO_4 \cdot 6H_2O$ | 25 | Ni source |
| $Na_2WO_4 \cdot H_2O$ | 50 | W source |
| $Na_3C_6H_5O_7 \cdot 2H_2O$ (Trisodium citrate) | 45 | Complexing agent |
| $C_6H_8O$ (Citric Acid) | 5.5 | Complexing agent |
| $C_7H_5NO_3S$ (Saccharin) | 0.8 | Softener |
| **Parameters** | **Value** | |
| Temperature (°C) | 65 | |
| Current density (A/dm$^2$) | 1,2,3,4,5 | |
| pH | 8 | |
| Cathode | Q235 | |
| Anode | Stainless steel | |
| Stirring speed (rpm) | 400 | |
| Heat treatment temperature (°C) | 400,600,900 | |

The surface morphology and microstructure of Ni-W alloys were studied using the scanning electron microscope (SEM FEI Quanta 200, Hillsboro, OR, USA) and transmission electron microscope (TEM JEOL JEM-2100, Tokyo, Japan). The composition of the coating was analyzed using an X-ray energy dispersive spectrometer (EDS Oxford Instruments X-Max N 80, Oxford, UK), while the Ni-W alloy coating was analyzed using X-ray diffraction (XRD Bruker D8 Advance, Shanghai, China) with Cu Kα radiation wavelength, a 2θ angle range of 30° to 100°, and a scanning speed of 3°/min. The energy changes and reaction of the coating in response to the changes in temperature were measured using differential scanning calorimetry (DSC TA Instruments Q200, New Castle, PA, USA). The heating rate was 10 °C/min. The electrocatalytic stability of Ni-W alloy coatings was tested using an HXS-1000A(Shanghai, China) microhardness tester under a 50 g load by monitoring the electrode reaction for 10 s. The average five-point values were obtained as the microhardness of the coating. The wear and friction coefficient (COF) of Ni-W samples was evaluated using ball-disk friction and a wear tester(UT-3000 AEP, Hillsboro, OR, USA), and $Al_2O_3$ balls with a diameter of 3 mm were the grinding materials for the coating. The wear tests were conducted under dry conditions (25 °C, 35% relative humidity, no lubricant) at a 5 N load and 5 cm/s sliding speed. Sample mass loss was calculated by weighing the mass of the sample before and after the friction and wear testing using a thermal analytical balance.

Potentiodynamic polarization and electrochemical impedance spectroscopy (EIS) were used to determine the corrosion resistance of the coating before and after heat treatment. Measurements were performed in 5wt.%NaCl solution using a three-electrode system and an electrochemical system (Autolab PGSTAT302N, Herisau, Switzerland). The platinum electrode was used as the auxiliary electrode, and the saturated calomel electrode (SCE) was used as the reference electrode. The potentiodynamic curve was recorded in the potential range of an open-circuit potential ±100 mV to achieve a rate of av = 1 mV/s. Electrochemical impedance spectroscopy measurements were performed for the anode, the cathode, and the AC signal at an open-circuit potential. The amplitude of AC signal used in these measurements was 10 mV. The frequency range of 100 kHz to 0.01 Hz covered 12 points per 10 octaves, and all electrochemical tests were carried out at 25 °C.

## 3. Results and Discussion

### 3.1. Surface Morphology and Composition

Figure 1 shows the Ni-W alloy coating micro-morphology at different current densities and heat treatment temperatures. The surface morphology of as-deposited Ni-W alloy coating depends on the current density. For example, the crystal of the coating is needle-like when the current density is 1 A/dm². However, the crystal of the coating changes from being needle-like to being a mixture of needle-like and cellular crystals when the current density increases to 2 A/dm². The crystallization changes completely to cellular crystal when the current density exceeds 3 A/dm². The cellular crystal is a cluster composed of fine grains grown or formed in a certain group or cluster. Moreover, the cluster increases with a high current density since the low current density benefits the slow discharge of metal ions in the cathode [21]. Therefore, the grain growth rate is faster than the nucleation rate of new grains. Generally, the high current density promotes grain refinement. Increasing the current density leads to a higher over potential which increases the nucleation rate, leading to the needle-like transformation of grains into cell assembly. These results are consistent with the results of Popczyk [22]. The high current density motivated a strong hydrogen evolution reaction, which increased the surface roughness of the coating. The W content in Ni-W alloy coating increased from 23 wt.% at 1 A/dm² to 44.9 wt.% at 5 A/dm² with high current density (Figure 2). The Ni-W alloy coating was completely amorphous when the W content was more than 40 wt.%.

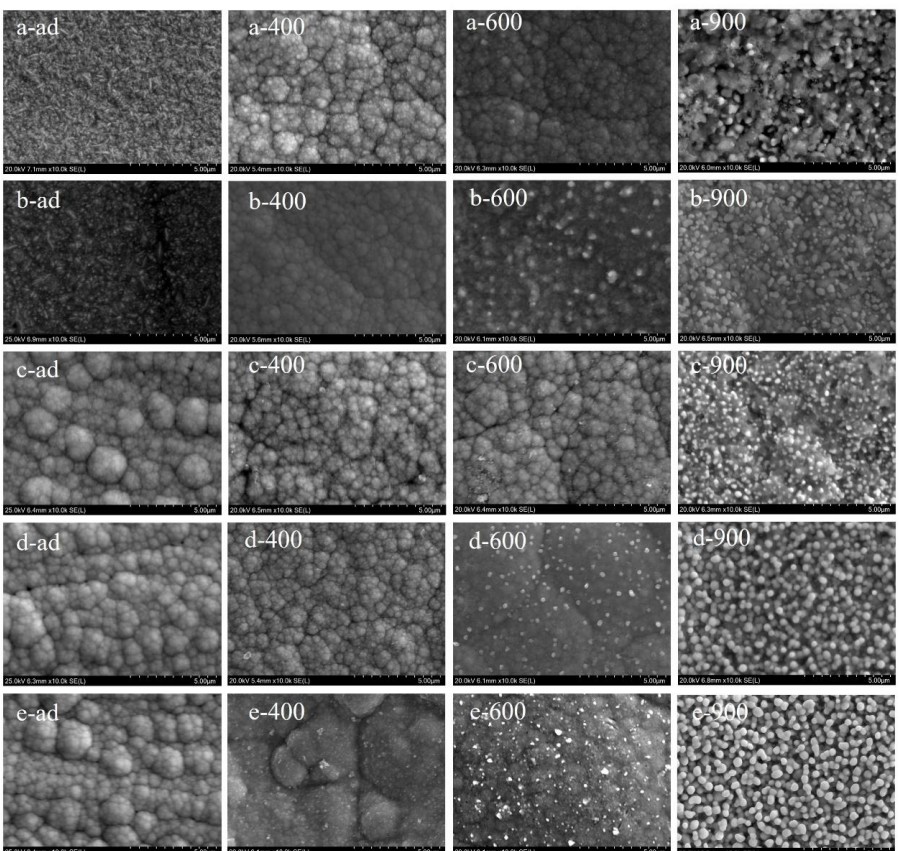

**Figure 1.** The Ni-W alloy coating micro-morphology under different current densities (**a**—1 A/dm$^2$, **b**—2 A/dm$^2$, **c**—3 A/dm$^2$, **d**—4 A/dm$^2$, **e**—5 A/dm$^2$) and heat treatment temperature (as-deposited 400–400 °C, 600–600 °C, and 900–900 °C).

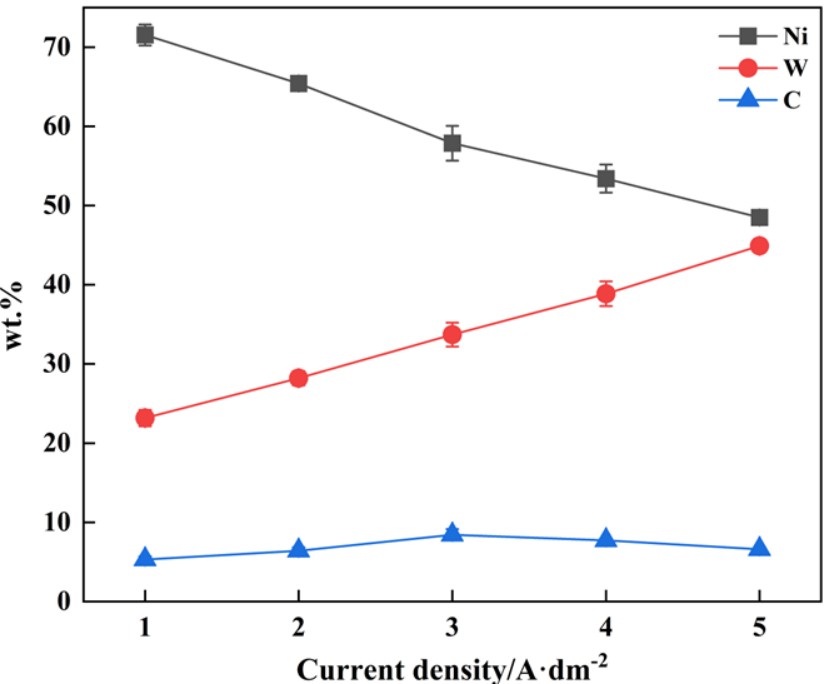

**Figure 2.** Element composition of Ni−W alloy coating at different current densities.

Therefore, the surface morphology of the cellular crystals in the shape of broccoli clusters with high current density [23]. Figure 1 shows the coating morphology with

acicular crystals at low current density were replaced by cellular crystals after heat treatment at 400 °C, indicating that the solid solution of W in Ni was more uniform under 400 °C heat treatment. In addition, the square particles precipitated in 5 A/dm² (Figure 1e-400). The coating surface was smooth and no longer showed the surface morphology of cellular crystals after heat treatment at 600 °C, increasing with a high current density. Spherical particles precipitated at 2−5A/dm². Great changes occurred in the surface morphology of Ni-W alloy coating at various current densities after heat treatment at 900 °C, and several spherical particles precipitated on the coating surface.

Figure 3 shows the mapping of the Ni-W alloy coating at different treatment temperatures. The distribution of elements on the surface of the coating was still uniform after heat treatment at 400 °C. The uniform distribution of Ni and W elements in the coating decreased, and the carbon elements began to aggregate in a small range under high temperatures, indicating that the originally uniformly dispersed carbon elements reacted with metals to form carbides. The W-rich and Ni-rich phases were formed on the surface of the coating at 900 °C. The carbon elements were concentrated in the position of the W-rich phase, indicating the possibility of forming Ni-W-C or intermetallic compounds of the W-C system. As shown in Figure 4, the analysis of the precipitated phase on the coating surface using EDS revealed a cubic crystal $Ni_4W$ at 400 °C. According to the description of the Ni-W alloy phase diagram [8], it is possible to yield $Ni_4W$ after heat treatment when the W content in Ni-W alloy coating is 28 wt.%. The precipitated phase of $Ni_6W_6C$ in the heat-treated sample at 600 °C was cubic crystal, which is consistent with the research results of Marvel [7]. Some carbon impurities are inevitably added to the coating due to organic complexing agents such as citrate. The precipitated phase of $Ni_6W_6C$ is often ignored. The hexagonal WC precipitated on the coating surface after heat treatment at 900 °C for two hours, growing into particles of 50–500 nm size during the heat treatment and distributed uniformly in the coating similar to that of the composite coating.

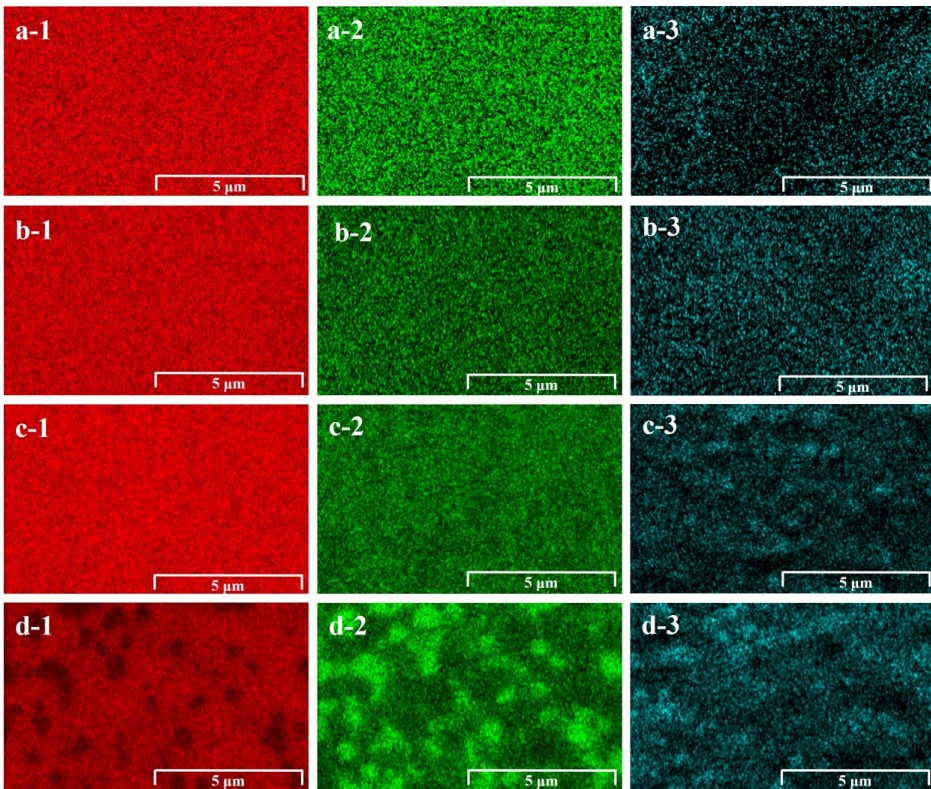

**Figure 3.** The mapping of Ni-W alloy coating with heat treatment at different temperature (**a**—400 °C, **b**—600 °C, **c**—900 °C), showing different elements (**1**—nickel element, **2**—tungsten element, **3**—carbon element).

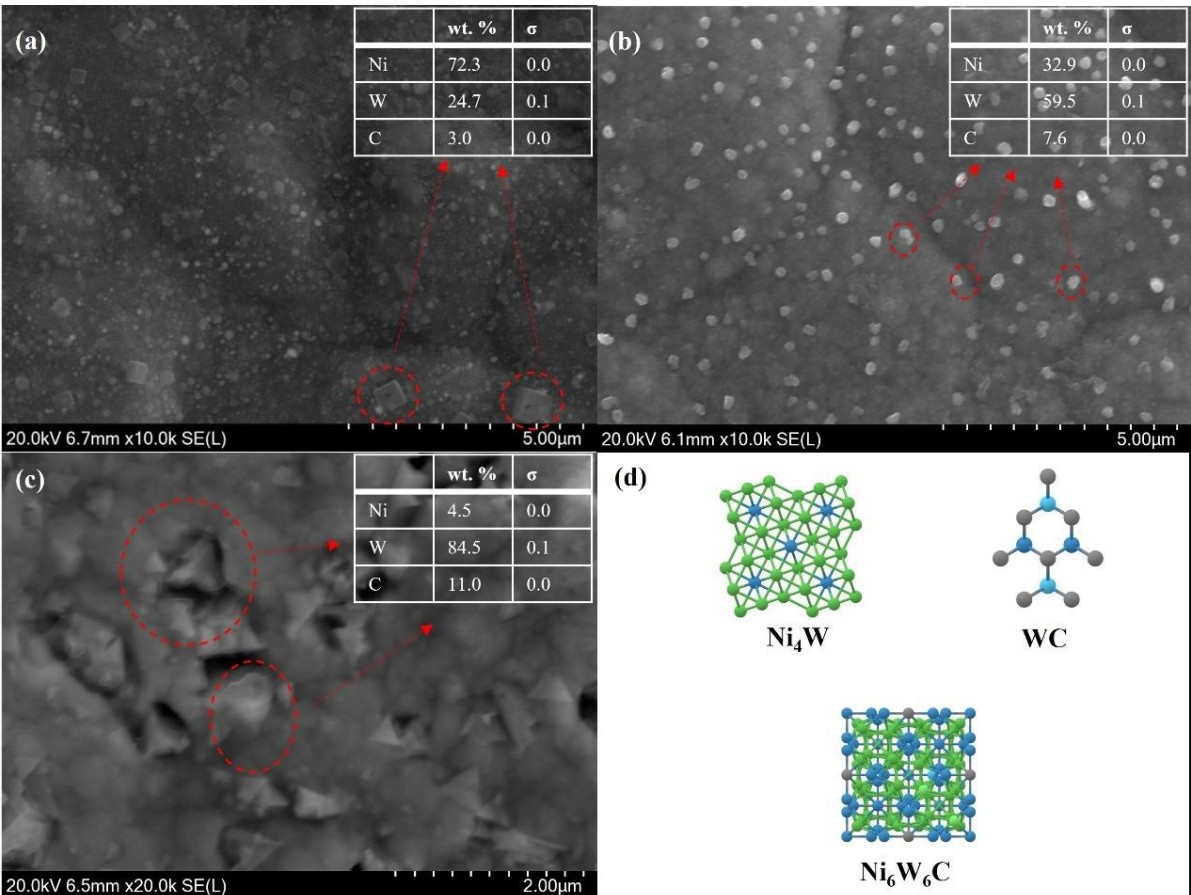

**Figure 4.** Intermetallic compounds precipitated by heat treatment at 400 °C (**a**), 600 °C (**b**), 900 °C (**c**), and schematic diagram of the crystal structure of $Ni_4W$, $Ni_6W_6C$, and WC (**d**).

### 3.2. Phase Analysis

The phase composition of the as-deposited Ni-W alloy cladding was found to be independent of current density. Figure 5a shows the XRD spectra of the electrodeposited Ni-W alloy at different current densities. The Ni (W) solid solution and Ni (fcc) had similar characteristic peaks. The Ni diffraction peak was sharp at low current density. The diffraction peak broadened, and its intensity decreased with the high current density. Thus, the grain size of the coating gradually decreased and transitioned to an amorphous state [24,25]. Due to the addition of W atoms with a larger atomic radius in the Ni matrix, the characteristic peak of the Ni (W) solid solution shifted to a lower angle compared with the standard peak of Ni (fcc), resulting in a larger lattice constant. Figure 5b–d shows the XRD patterns of Ni-W alloy coatings with different current densities after heat treatment at 400 °C, 600 °C, and 900 °C, respectively. The coating revealed a sharper peak after heat treatment at 400 °C, indicating the coating crystallization, especially at 600 °C and 900 °C. The characteristic peak of the Ni (W) solid solution gradually shifted to a high angle with the high heat treatment temperature, finally approaching the standard peak of Ni (fcc). Thus, the segregation of the W element in the coating increased with a high heat treatment temperature. In addition, high heat treatment temperature yielded $Ni_4W$, $Ni_6W_6C$, and WC in XRD patterns, which was consistent with the results in Figure 4. Moreover, the precipitated phase was related to the heat treatment temperature and not the current density. However, the deposited layer of 1 A/dm$^2$ yielded no WC peak after heat treatment at 900 °C, and the W content was too low.

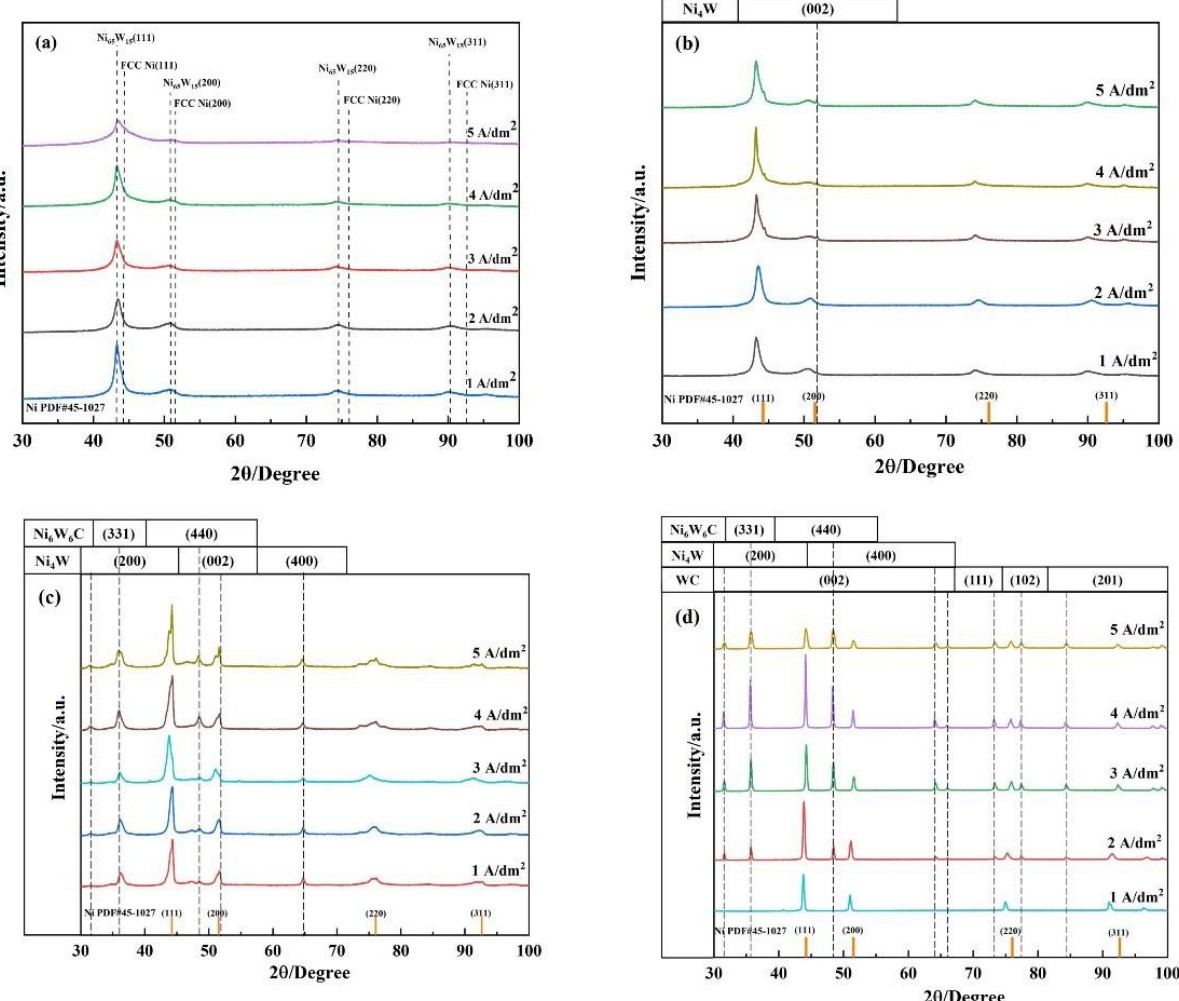

**Figure 5.** XRD patterns of coatings (exfoliated phase) at different current densities before heat treatment (**a**) and after heat treatment at 400 °C (**b**), 600 °C (**c**), and 900 °C (**d**).

Figure 6 shows the grain size of each coating phase calculated before and after heat treatment using the Scherrer formula. High currency density reduced the grain size of the as-deposited layer. The (111) crystal plane of the coating increased a little in their grain size after heat treatment at 400 °C, all of which were nanocrystals of 10nm, and the growth slope increased with high current density. The as-deposited samples with different current densities were analyzed using DSC to explain this phenomenon further since the (111) crystal plane of the Ni-W alloy coating is dominant. As shown in Figure 7, endothermic peaks exist in the thermal spectra of all coatings. The overall reaction was endothermic. Thus, the coating crystallized, and the grains grew continuously at high temperatures. The first endothermic peak for all coatings was in the range of 145.8–154.5 °C. Several organic complexing agents enter the coating during the deposition process and are thus regarded as the decomposition of organic matter [26]

Crystallization begins at the second endothermic peak (337.3–414.5 °C), and the peak shifts to the left under high current density, indicating that the phase transition temperature decreases with high current density. This may be because the W content increases with high current density. High W content promotes grain boundary segregation during heat treatment, which in turn reduces the grain boundary energy $\gamma$. The lower grain boundary energy $\gamma$ promotes the grain growth of Ni (W) solid solution [27,28], which is in agreement with the results of Figure 6. However, a too-high W content affects the thermal stability of the Ni-W alloy. Following heat treatment at 600°C, the grain size of the Ni (W) solid

solution was observed to increase, further increasing to 20–50 nm after heat treatment at 900 °C, yet the coating was still nanocrystalline. High heat treatment also increases the grain size of $Ni_4W$ and $Ni_6W_6C$. WC precipitates only at 900 °C, and the grain size of the deposited layer with different current densities is 25–30 nm.

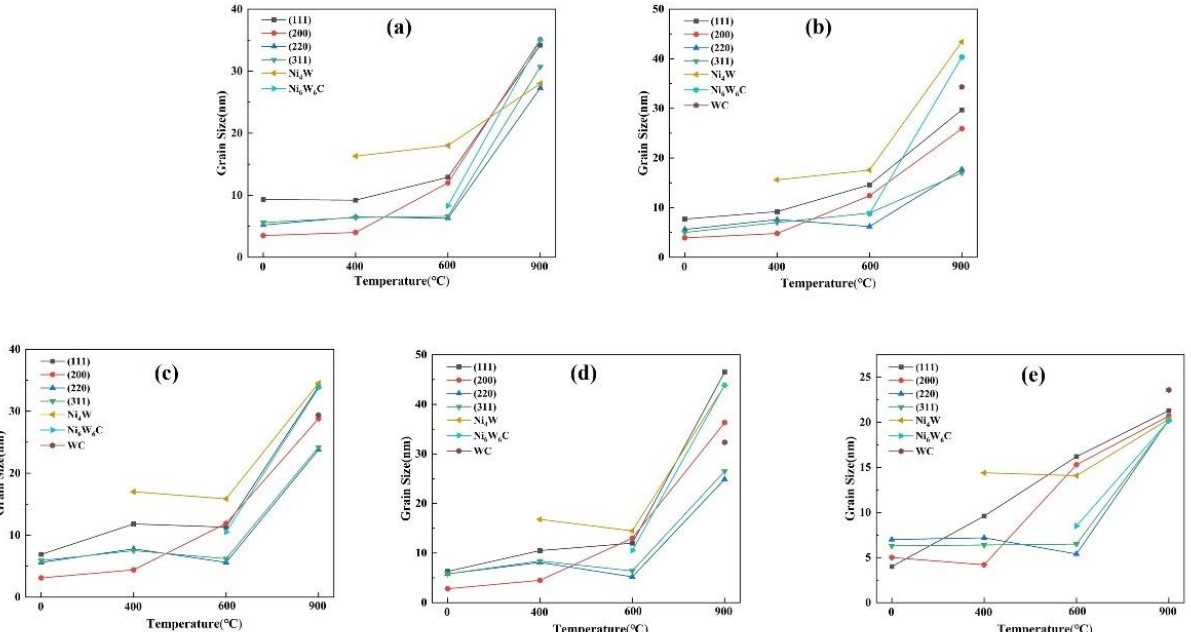

**Figure 6.** The effects of heat treatment temperature on grain size of various phases in Ni-W alloy coatings at $1A/dm^2$ (**a**), $2A/dm^2$ (**b**), $3A/dm^2$ (**c**), $4A/dm^2$ (**d**), and $5A/dm^2$ (**e**).

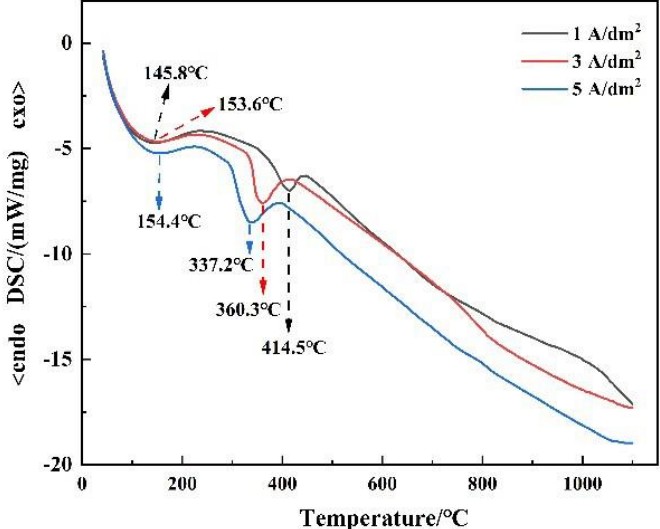

**Figure 7.** DSC thermal analysis diagram of Ni-W alloy coating prepared with different current densities.

The phase proportion of the deposit before and after heat treatment was further analyzed using the semi-quantitative analysis of the RIR value [29] as shown in Formula (1):

$$RIR_A = I_A/I_{col} \tag{1}$$

$I_A$ and $I_{col}$ are the integral strengths of the strongest peaks of phase $A$ and reference material ($\alpha$-Al$_2$O$_3$), respectively. In the case of $N$ phases in a system, the appropriate fraction of phase $i$ is given by the *RIR* value of each phase as shown in Formula (2):

$$W_i = \frac{I_i / RIR_i}{\sum_{i=1}^{N} I_i / RIR_i} \tag{2}$$

As shown in Figure 8, the proportion of the (111) crystal plane of the deposition layer decreased with high heat treatment temperature and was not affected by current density. The phase proportion of other crystal planes of Ni (W) solid solution fluctuates, with the (200) crystal plane fluctuating the most. The proportion of Ni$_4$W in the coating is low, not exceeding 5 wt.% at different heat treatment temperatures, indicating that Ni$_6$W$_6$C is preferentially precipitated at higher heat treatment temperatures. The Ni$_6$W$_6$C precipitation is highest at 600 °C when the current density is 1–2 A/dm$^2$ and highest at 900 °C when the current density is 3–5A/dm$^2$. Therefore, Ni$_6$W$_6$C formation is more favorable under high tungsten. Otherwise, the formation reaction reaches equilibrium at a high temperature. WC was produced, and its proportion increased from 0% (1 A/dm$^2$) to 5%(5 A/dm$^2$) only during heat treatment at 900 °C. Therefore, a high W content in Ni-W alloy coating makes it easier to form WC.

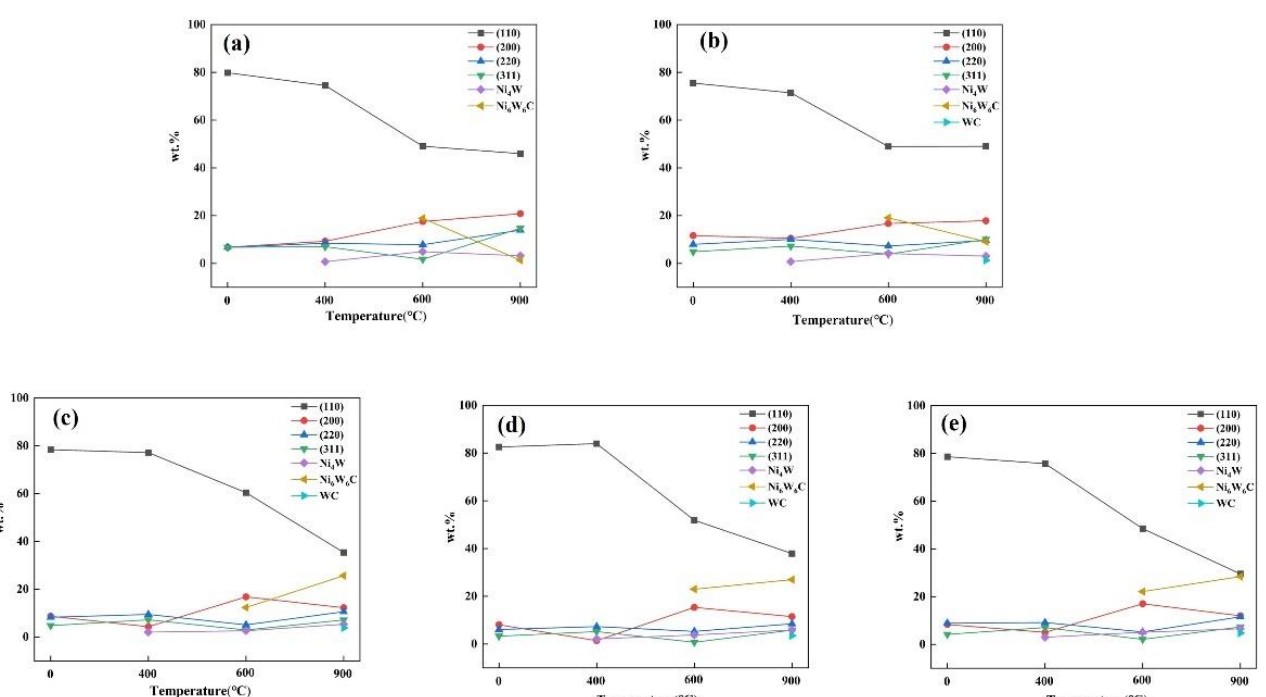

**Figure 8.** The effects of heat treatment temperature on the mass percentage of each phase in Ni-W alloy coating at current densities of 1A/dm$^2$ (**a**), 2A/dm$^2$ (**b**), 3A/dm$^2$ (**c**), 4A/dm$^2$ (**d**), and 5A/dm$^2$ (**e**).

Figure 8 shows that the content of each crystal plane in Ni (W) solid solution varied with the heat treatment temperature. We calculated the relative texture coefficient for each crystalline plane of the coating using Formula (3) [30] to further quantify the change of each crystal plane with the heat treatment temperature, characterizing the relative degree of preferred orientation between all the crystal planes.

$$RTC_{(hkl)} = \frac{R_{(hkl)}}{\sum_1^n R_{(hkl)}} \times 100\%; \; R_{(hkl)} = \frac{I_{s(hkl)}}{I_{p(hkl)}} \tag{3}$$

where $I_s$ *(hkl)* and $I_p$ *(hkl)* are the diffraction intensities of the *(hkl)* plane measured by the deposition layer and the standard sample of Ni powder (JCPDSno.04–0850), respectively. Table 2 shows that the Ni-W alloy coating had a preferred orientation in the as-deposited state, and the RTC (111) decreased after heat treatment at 400 °C. However, the crystal plane still had a growth advantage. The growth of the (111) crystal plane was further inhibited, while the growth of the (200), (220), and (311) crystal planes was promoted, especially for the (200) crystal plane, when the heat treatment temperature was 600 °C. The (111) and (200) crystal planes were dominant growth planes. The relative texture coefficients of each crystal plane of the coating were average and without the preferred orientation after heat treatment at 900 °C. Therefore, the change of RTC was greatly affected by temperature and not current density.

**Table 2.** The effects of heat treatment temperature on the relative texture coefficient of each crystal plane of Ni-W alloy coating.

| Current Density (A/dm$^2$) | RTC | | | |
|---|---|---|---|---|
| | **(111)** | **(200)** | **(220)** | **(311)** |
| As-deposited | | | | |
| 1 | 0.501 | 0.152 | 0.210 | 0.135 |
| 2 | 0.414 | 0.002 | 0.002 | 0.001 |
| 3 | 0.479 | 0.001 | 0.002 | 0.001 |
| 4 | 0.597 | 0.001 | 0.002 | 0.001 |
| 5 | 0.574 | 0.001 | 0.002 | 0.001 |
| 400 | | | | |
| 1 | 0.450 | 0.108 | 0.247 | 0.194 |
| 2 | 0.436 | 0.150 | 0.243 | 0.170 |
| 3 | 0.476 | 0.085 | 0.257 | 0.181 |
| 4 | 0.580 | 0.081 | 0.207 | 0.131 |
| 5 | 0.609 | 0.094 | 0.174 | 0.122 |
| 600 | | | | |
| 1 | 0.317 | 0.243 | 0.239 | 0.200 |
| 2 | 0.328 | 0.285 | 0.225 | 0.160 |
| 3 | 0.458 | 0.303 | 0.151 | 0.087 |
| 4 | 0.455 | 0.178 | 0.208 | 0.157 |
| 5 | 0.380 | 0.277 | 0.203 | 0.138 |
| 900 | | | | |
| 1 | 0.199 | 0.215 | 0.301 | 0.284 |
| 2 | 0.281 | 0.231 | 0.252 | 0.235 |
| 3 | 0.367 | 0.274 | 0.187 | 0.171 |
| 4 | 0.313 | 0.221 | 0.258 | 0.208 |
| 5 | 0.249 | 0.223 | 0.292 | 0.236 |

*3.3. Mechanical Properties*

The hardness of the Ni-W alloy coating decreased with large grain size, which accords with the Hall–Petch relation. That is, small grain size increases grain boundary volume, hindering the movement of dislocations and increasing the coating hardness [31,32]. Figure 9a shows that the as-deposited sample tally with the Hall–Petch relation in the range of 1–4A/dm$^2$. In addition, the coating hardness decreased at 5 A/dm$^2$ due to the transition of the coating to nanocrystalline–amorphous at high current density. Studies have revealed that when the grain size of Ni-W alloy coating is smaller than that of 10 nm, the hardness no longer increases with small grain size, and there is no tally with the Hall–Petch relation [6]. The coating hardness greatly increased after heat treatment at 400 °C.

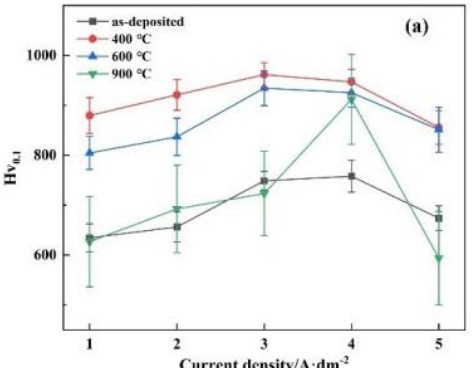
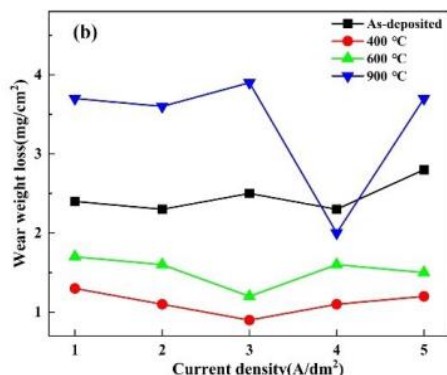

**Figure 9.** Mechanical properties of Ni-W alloy coating at different heat treatment temperatures (**a**) hardness; (**b**) weight loss after friction and wear.

Moreover, the grain size does not change much after heat treatment at 400 °C. Therefore, grain size is unlikely to be the key to changing hardness. Ni4W may be uniformly distributed in the coating after precipitation. On the one hand, Ni4W plays the role of pinning, restraining grain growth, while on the other hand, it plays the effect of dispersion strengthening similar to that of the composite coating. Some studies attribute this phenomenon to the relaxation of grain boundaries, which annihilates the excess dislocations at the grain boundaries, increasing the critical shear stress needed to trigger slip [33]. Heat treatment at 600 °C also increased coating hardness, but not as good as at 400 °C due to the large grain size. The coating hardness after heat treatment at 900 °C was similar to that of the as-deposited coating due to the serious W segregation at this temperature and the formation of two W- rich and Ni-rich phases, loosening the coating. In addition, sample hardness increased abnormally after heat treatment at 4 A/dm$^2$ due to the formation of hard phase particles, such as WC on the coating surface (Figure 4b), strengthening the coating surface and improving the coating hardness. Although the wear resistance in Figure 9b was not strictly proportional to the coating hardness, it was consistent with the trend in the change of hardness. The wear resistance of the sample was best at 400 °C heat treatment, followed by 600 °C, as-deposited, and 900 °C. The wear resistance was stable with the change in current density. The coating hardness and the precipitation of Ni$_4$W, Ni$_6$W$_6$C, and WC as a result of heat treatment influences the wear resistance of the coating. Humam [16] suggested that doping WC and TaC into the Ni-W alloy coating can make the coating non-porous and compact, improving its mechanical properties. At the same time, the hard phase precipitated by heat treatment is more evenly distributed in the coating, and controlling a certain temperature also improves the coating compactness [34]. As shown in Figure 10, sample thickness of the atomic interdiffusion layer at the interface of the as-deposited sample and at 400 °C was about 500 nm, with negligible effects, while the interdiffusion layers of the samples at 600 °C and 900 °C heat treatment were 10 μm and 20 μm, respectively. Therefore, Ni-W-Fe-C quaternary system did not form inside the coating. Thus, the composition of the coating cannot be controlled after heat treatment. These findings also explain why hard phase particles, such as WC, are precipitated after heat treatment at 900 °C without the improvement of mechanical properties.

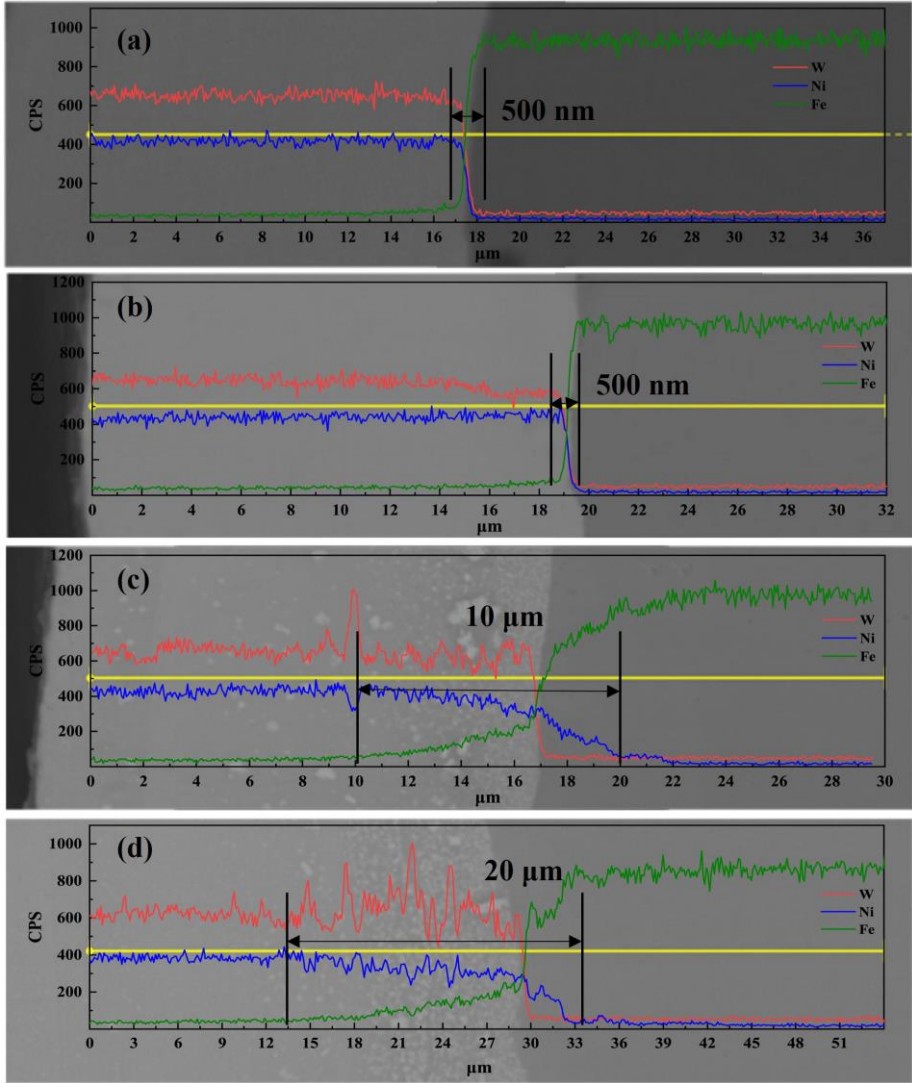

**Figure 10.** Line scanning diagram of Ni-W alloy–substrate interface before heat treatment (**a**) and after heat treatment at 400 °C (**b**), 600 °C (**c**), and 900 °C (**d**).

*3.4. Corrosion Resistance*

Figure 11 shows the polarization curves of plated and heat-treated Ni-W coatings. Electrochemical corrosion parameters derived using the Tafel extrapolation are given in Tables 3 and 4 [35]. Higher current densities result in a denser and more uniform coating structure, reducing the likelihood of corrosive substances permeating through defects or gaps in the coating. Additionally, higher current densities can induce a transformation of the coating into an amorphous state, consequently enhancing corrosion resistance. The as-deposited sample of 5 A/dm$^2$ had the lowest corrosion current ($i_{corr}$), indicating that the coating was nanocrystalline–amorphous at high current density with excellent corrosion resistance. The samples heat treated at 400 °C had better corrosion resistance than those with higher corrosion potential ($E_{corr}$). However, the samples with different current densities narrowly differed. The corrosion resistance of the samples heat treated at 600 °C was improved, while those heat treated at 900 °C had the worst corrosion resistance. Figure 12 shows the AC impedance (EIS) spectra of the samples before and after heat treatment at different temperatures. Nyquist diagram revealed an arc with different radii, and the only time constant of the impedance diagram was determined. Therefore, the equivalent circuit diagram (ECD) (Figure 13) was drawn to calculate the corrosion parameters, where $R_s$ is the resistance of the solution, CPE is the electrical double-layer

capacitance, and $R_{ct}$ is the charge transfer resistance at the coating/substrate interface. The fitting data are shown in Table 3 [36]. The $R_{ct}$ is related to corrosion resistance. The corrosion resistance of the coating increases with a high $R_{ct}$ value. The result of EIS was similar to that of the polarization curve, and the samples heat treated at 400 °C and 600 °C showed better corrosion resistance, attributed to the precipitates, such as $Ni_4W$ and $Ni_6W_6C$, making the coating denser. Long et al. [37] revealed that the corrosion resistance of the Ni-W alloy is affected by the boundary of cellular crystal clusters. The corrosion resistance of the coating worsens with a high density of intercluster boundary (DIB). As shown in Figure 1, the boundary of cellular crystal clusters was eliminated under heat treatment at 400 °C and 600 °C, making the coating surface more compact. The surface of the coating heat treated at 600 °C was smoother than at 400 °C, hence the best corrosion resistance. It is also possible that the crystallization orientation of Ni (W) solid solution was changed by heat treatment at 600 °C, and the promoted (200) crystal plane played a role in corrosion resistance. The surface porosity of the coating heat treated at 900 °C was too large due to the severe segregation of W and the formation of many precipitates, reducing the corrosion resistance. It is also possible that the composition of the coating was changed due to the interdiffusion between the coating and the substrate.

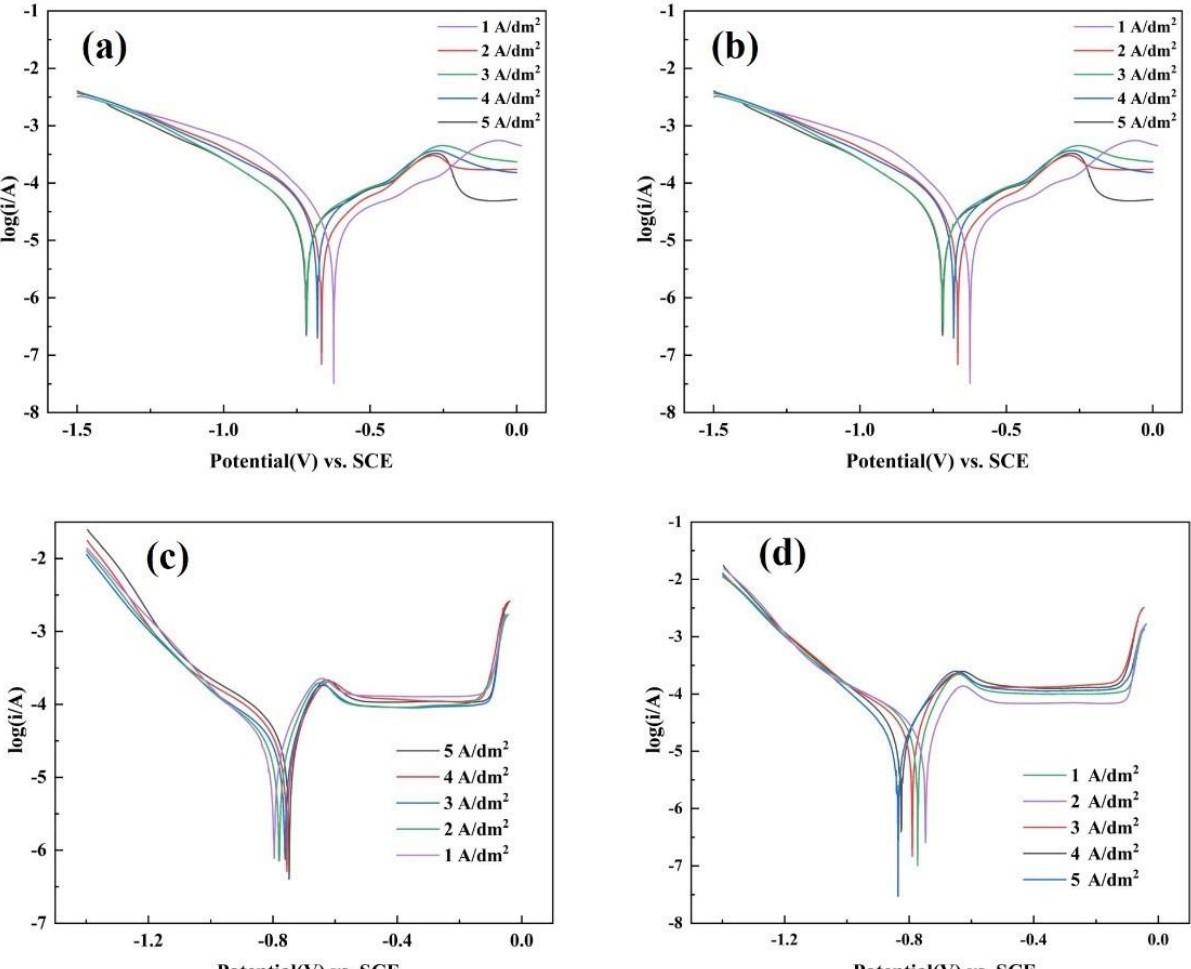

**Figure 11.** Tafel polarization curves of samples before heat treatment (**a**) and heat treated at 400 °C (**b**), 600 °C (**c**), and 900 °C (**d**).

**Table 3.** Electrochemical data obtained from polarization curve.

|  | $E_{corr}$ (V) (V vs.SCE) | $I_{corr}$(A) (A) |
|---|---|---|
| As-deposited |  |  |
| $1A/dm^2$ | $-0.71$ | $2.85 \times 10^{-4}$ |
| $2 A/dm^2$ | $-0.68$ | $2.63 \times 10^{-4}$ |
| $3 A/dm^2$ | $-0.67$ | $2.32 \times 10^{-4}$ |
| $4 A/dm^2$ | $-0.67$ | $2.14 \times 10^{-4}$ |
| $5 A/dm^2$ | $-0.65$ | $1.70 \times 10^{-4}$ |
| 400 °C |  |  |
| $1A/dm^2$ | $-0.62$ | $4.36 \times 10^{-6}$ |
| $2 A/dm^2$ | $-0.66$ | $4.58 \times 10^{-6}$ |
| $3 A/dm^2$ | $-0.72$ | $9.21 \times 10^{-6}$ |
| $4 A/dm^2$ | $-0.68$ | $8.43 \times 10^{-6}$ |
| $5 A/dm^2$ | $-0.72$ | $9.86 \times 10^{-6}$ |
| 600 °C |  |  |
| $1A/dm^2$ | $-0.79$ | $6.35 \times 10^{-6}$ |
| $2 A/dm^2$ | $-0.78$ | $3.13 \times 10^{-6}$ |
| $3 A/dm^2$ | $-0.76$ | $2.64 \times 10^{-6}$ |
| $4 A/dm^2$ | $-0.76$ | $2.11 \times 10^{-6}$ |
| $5 A/dm^2$ | $-0.75$ | $1.96 \times 10^{-6}$ |
| 900 °C |  |  |
| $1A/dm^2$ | $-0.77$ | $2.10 \times 10^{-5}$ |
| $2 A/dm^2$ | $-0.75$ | $1.33 \times 10^{-5}$ |
| $3 A/dm^2$ | $-0.79$ | $5.10 \times 10^{-5}$ |
| $4 A/dm^2$ | $-0.82$ | $1.81 \times 10^{-5}$ |
| $5 A/dm^2$ | $-0.84$ | $3.72 \times 10^{-5}$ |

**Table 4.** Electrochemical data obtained from EIS spectrum.

|  | Rs($\Omega$) ($\Omega \cdot cm^{-2}$) | CPE-$Q_{dl}$ ($\Omega^{-1}s^n/cm^2$) $(cm^2) \times 10^{-6}$(S sec$^n$) | $R_{ct}$ ($\Omega \cdot cm^{-2}$) |
|---|---|---|---|
| As-deposited |  |  |  |
| $1 A/dm^2$ | 26.57 | $2.076 \times 10^{-4}$ | 533.7 |
| $2 A/dm^2$ | 25.32 | $1.452 \times 10^{-4}$ | 856.1 |
| $3 A/dm^2$ | 24.13 | $9.193 \times 10^{-5}$ | 1120.5 |
| $4 A/dm^2$ | 28.32 | $9.953 \times 10^{-5}$ | 965.5 |
| $5 A/dm^2$ | 24.61 | $8.593 \times 10^{-5}$ | 1882.4 |
| 400 °C |  |  |  |
| $1 A/dm^2$ | 21.37 | $1.09 \times 10^{-4}$ | 2104.1 |
| $2 A/dm^2$ | 21.96 | $8.59 \times 10^{-4}$ | 1928.1 |
| $3 A/dm^2$ | 21.36 | $9.693 \times 10^{-5}$ | 1526.4 |
| $4 A/dm^2$ | 21.63 | $3.083 \times 10^{-5}$ | 1598.4 |
| $5 A/dm^2$ | 22.22 | $2.683 \times 10^{-6}$ | 1342.4 |
| 600 °C |  |  |  |
| $1A/dm^2$ | 27.84 | $9.875 \times 10^{-5}$ | 2375.1 |
| $2 A/dm^2$ | 24.36 | $9.365 \times 10^{-5}$ | 2472.6 |
| $3 A/dm^2$ | 25.63 | $8.365 \times 10^{-5}$ | 2672.2 |
| $4 A/dm^2$ | 26.96 | $7.635 \times 10^{-5}$ | 2885.7 |
| $5 A/dm^2$ | 24.58 | $7.132 \times 10^{-5}$ | 3031.4 |
| 900 °C |  |  |  |
| $1A/dm^2$ | 27.32 | $2.324 \times 10^{-5}$ | 863.5 |
| $2 A/dm^2$ | 26.32 | $1.124 \times 10^{-5}$ | 1130.2 |
| $3 A/dm^2$ | 24.21 | $4.325 \times 10^{-4}$ | 436.2 |
| $4 A/dm^2$ | 24.32 | $3.241 \times 10^{-4}$ | 432.1 |
| $5 A/dm^2$ | 23.36 | $1.103 \times 10^{-4}$ | 536.2 |

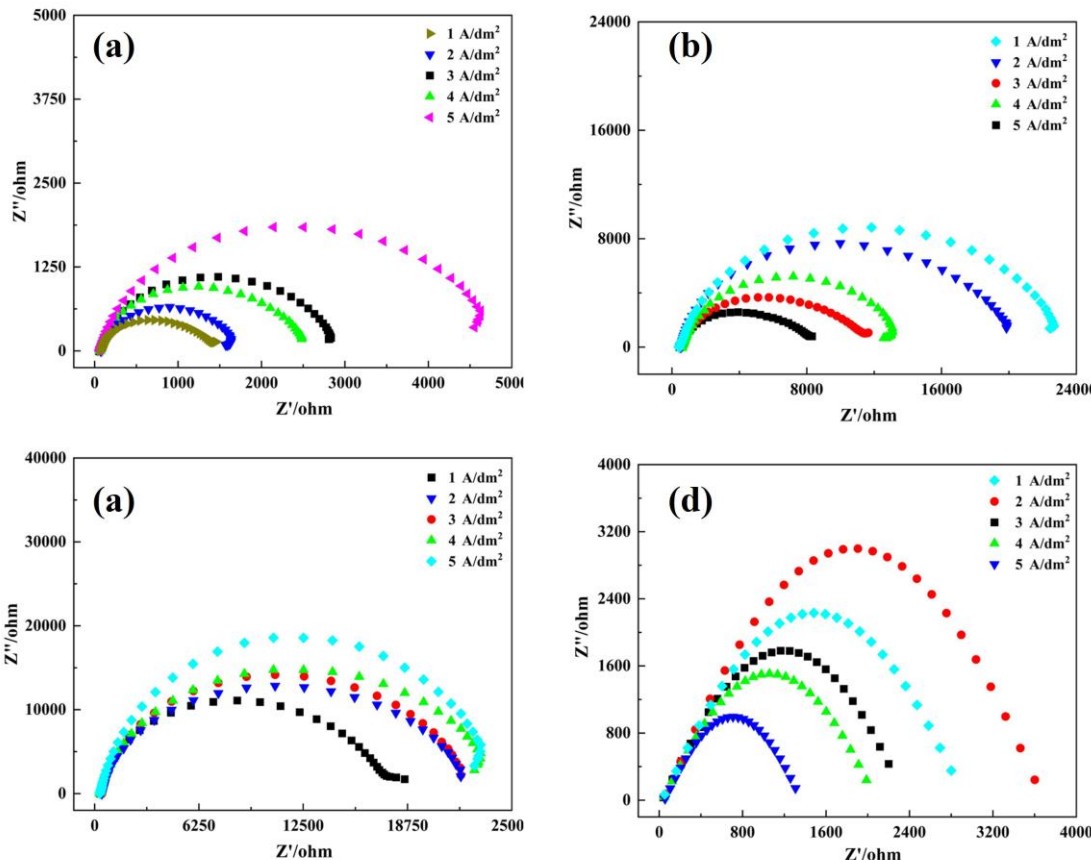

**Figure 12.** EIS curve of samples before heat treatment (**a**) and heat treated at 400 °C (**b**), 600 °C (**c**), and 900 °C (**d**).

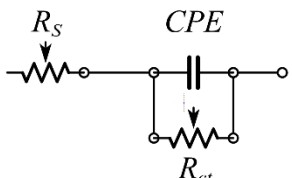

**Figure 13.** Equivalent circuit of impedance measurement.

## 4. Conclusions

In this study, Ni-W alloy coatings prepared under different current densities were heat treated at 400 °C, 600 °C, and 900 °C. By comparing their phase distribution, mechanical properties, and corrosion behavior, the following conclusions have been drawn.

The composite structure of nanocrystal and amorphous in Ni-W alloy coating crystallizes, and the grain size increases after heat treatment above 400 °C. $Ni_4W$ forms at 400 °C, and $Ni_6W_6C$ forms at 600 °C when W reaches a certain content. Heat treatment inhibits the preferred orientation of the (111) crystal plane. The coating hardness and wear resistance are highest at 400 °C heat treatment. Due to its amorphous structure, the as-plated sample at 5 $A/dm^2$ exhibits the best corrosion resistance. As the heat treatment temperature increases, the coating structure becomes denser, resulting in the corrosion resistance of the coating heat treated at 600 °C being superior to the other groups. Choosing the appropriate heat treatment temperature obtains the target hard particle precipitation phase, eliminates the boundary of cellular crystals on the surface of the coating, and makes the structure of the coating more compact. The WC hard particles are produced during heat treatment at 900 °C. However, the coating properties are not improved due to the interdiffusion between the coating and the substrate. Therefore, the heat treatment of Ni-W

alloy coating should consider not only the phase change of the coating but also the effect of interdiffusion with the substrate.

**Author Contributions:** Y.X. and D.W. provide ideas, M.S. provides funding support, H.W., R.G., T.Q. and S.H. complete other tasks. All authors have read and agreed to the published version of the manuscript.

**Funding:** This research was funded by National Natural Science Foundation of China (52174280) and Science and Technology Program of Suzhou (SYG202022).

**Institutional Review Board Statement:** Not applicable.

**Informed Consent Statement:** Not applicable.

**Data Availability Statement:** Not applicable.

**Conflicts of Interest:** The authors declare no conflict of interest.

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
