# Peer review of "The Effect of Heat Treatment on Phase Structure and Mechanical and Corrosion Resistance Properties of High Tungsten Ni-W Alloy Coating"

_coatings, doi:10.3390/coatings13091651_

Round 1
Reviewer 1 Report
This manuscript is devoted to studying the effect of heat treatment on phase and properties of high 2 tungsten Ni-W alloy coating. The manuscript is well-written and well organized and includes some merits and can be accepted for publication after incorporating following suggestions:
1. “This study investigated the surface morphology, phase composition……….” Rewrite this statement to make it more clear. (Line No 9-11)
2. Add some more quantitative results in the abstract section.
3. “Cemented carbide is a powder metallurgy produced……..” Rewrite this statement to make it more clear. (Line No 19-20)
4. “However, composite plating has several shortcomings, such as high equipment requirements and complex operation” Is this statement correct? Please verify. (Line No 49-50)
5. Were substates mechanically polished before plating process?
6. Was there any delamination observed at higher temperature heat treatment due to the difference in thermal expansion of coating and substrate?
7. “Besides, the cluster increases with high current density since the low current density”……….Provide suitable reference. (Line No 116)
8. In Fig 2, why C content is more at current density of 3 A/dm2?
9. How you confirmed uniform dispersion of elements at 400℃? (Line No 142)
10. Add corrosion results in conclusion section.
Minor editing of English language required
Author Response
Dear Editors and Reviewers:
Thank you for your letter and for the reviewers’ comments concerning our manuscript entitled “The effect of heat treatment on phase and properties of high tungsten Ni-W alloy coating” (coatings-2587350). Those comments are all valuable and very helpful for revising and improving our paper, as well as the important guiding significance to our research. We have studied comments carefully and have made correction which we hope to meet with approval. Revised portions are marked in red in the paper. The main corrections in the paper and the responds to the reviewer’s comments are as flowing:
Responds to the reviewer’s comments:
Reviewer #1:
1.Comment: “This study investigated the surface morphology, phase composition……….” Rewrite this statement to make it more clear. (Line No 9-11)
Response: I apologize for our unclear expression. We have now revised that section to“The present study investigated the surface morphology, phase composition, mechanical properties, and corrosion resistance of Ni-W alloy coatings prepared under current densities of 1-5A/dm², after undergoing heat treatment at 400, 600, and 900℃. ”
- Comment: Add some more quantitative results in the abstract section.
Response: We appreciate the valuable feedback from the reviewer. We have incorporated more quantitative results into the abstract section.
- Comment: “Cemented carbide is a powder metallurgy produced……..” Rewrite this statement to make it more clear. (Line No 19-20)
Response: I apologize for our unclear expression. We have now revised that section to Cemented carbide is a high-hardness, wear-resistant composite material made from tungsten carbide particles bonded with a metal binder like cobalt or nickel, primarily used in cutting tools and wear-resistant components.
- Comment: “However, composite plating has several shortcomings, such as high equipment requirements and complex operation” Is this statement correct? Please verify. (Line No 49-50)
Response: As suggested by the reviewer, indeed, this sentence was unclear. We have now revised it to However, composite plating has several drawbacks, such as high equipment requirements and complex operations. Additionally, increasing the quantity of composite particles in the coating is also challenging.
- Comment: Were substates mechanically polished before plating process?
Response: We selected a commercially available, clean, and bright Q235 steel sheet as our cathode material. Prior to electroplating, a simple degreasing and acid cleaning process was applied. Therefore, we did not perform any mechanical polishing on it.
- Comment: Was there any delamination observed at higher temperature heat treatment due to the difference in thermal expansion of coating and substrate?
Response: We appreciate the constructive questions from the reviewing experts. Due to the extensive hydrogen evolution reactions during the electroplating process, the coating experiences significant tensile stress. As the thermal expansion coefficient of nickel is greater than that of iron, the metal's thermal expansion during heat treatment mitigates the coating's tensile stress to a certain extent. This helps suppress issues such as cracking and delamination in the coating.
- Comment: “Besides, the cluster increases with high current density since the low current density”……….Provide suitable reference. (Line No 116)
Response: We appreciate the reviewer's reminder, and we have added the references in the manuscript. [21]Suvorov D V, Gololobov G P, Tarabrin D Y, et al. Electrochemical deposition of Ni–W crack-free coatings[J]. Coatings, 2018, 8(7): 233. https://doi.org/10.3390/coatings8070233
- Comment: In Fig 2, why C content is more at current density of 3 A/dm2?
Response: The main source of carbon in the coating is the doping of complexing agents and their decomposition products. With higher current density, there is a greater amount of decomposition products from the cathodic complexing agents, making it easier for them to be incorporated into the coating. As the current density increases, the nucleation rate also increases. This leads to less available space between grains to accommodate carbon doping. As a result, the carbon content initially increases and then decreases with increasing current density.
- Comment: How you confirmed uniform dispersion of elements at 400℃? (Line No 142)
Response: This can be observed from Figure 3. The distribution of carbon content in the as-plated state is not uniform (a-3), and when the heat treatment temperature exceeds 400℃, the distribution of nickel, tungsten, and carbon elements also becomes uneven.
- Comment: Add corrosion results in conclusion section.
Response: I apologize, we inadvertently omitted the conclusion regarding the corrosion aspect in the conclusion section. We have now added the necessary content to enhance the conclusion.
Special thanks to you for your good comments.

Reviewer 2 Report
- The title can be modified as ....The effect of heat treatment on phase structure and mechanical and corrosion resistance properties of high tungsten Ni-W alloy coating.
- Electrochemical data obtained from the polarization curves and EIS spectra can be shown in separate tables.
- There are several English mistakes. For example, in Line 36…avoid capital letter for Carbon. Line 36 … Use either C or carbon throughout. Line 38… Therefore, in this study, the common Ni-W alloy coating was Ni-W-C ternary alloy...This sentence is not clear. Line 127…of the is.
- Figures 6, 8 and 11 need to be clearly seen.
Need a moderate english check.
Author Response
Dear Editors and Reviewers:
Thank you for your letter and for the reviewers’ comments concerning our manuscript entitled “The effect of heat treatment on phase and properties of high tungsten Ni-W alloy coating” (coatings-2587350). Those comments are all valuable and very helpful for revising and improving our paper, as well as the important guiding significance to our research. We have studied comments carefully and have made correction which we hope to meet with approval. Revised portions are marked in red in the paper. The main corrections in the paper and the responds to the reviewer’s comments are as flowing:
Responds to the reviewer’s comments:
1.Comment: The title can be modified as ....The effect of heat treatment on phase structure and mechanical and corrosion resistance properties of high tungsten Ni-W alloy coating.
Response: We appreciate the reviewer's suggestion. Indeed, there were issues with the previous title. We have now replaced the title as per the reviewer's recommendation.
- Comment: Electrochemical data obtained from the polarization curves and EIS spectra can be shown in separate tables.
Response: As mentioned by the reviewer, Table 3 was indeed too bulky. We have now divided it into two separate tables.
- Comment: There are several English mistakes. For example, in Line 36…avoid capital letter for Carbon. Line 36 … Use either C or carbon throughout. Line 38… Therefore, in this study, the common Ni-W alloy coating was Ni-W-C ternary alloy...This sentence is not clear. Line 127…of the is.
Response: We appreciate the reviewer's corrections. We have made revisions in the original text addressing the issues raised by the reviewer.
The unclearly worded sentence has been revised to“Therefore, in this study, the Ni-W alloy coating is considered a ternary alloy, and carbon element can react with nickel and tungsten elements and precipitate under certain heat treatment conditions”
- Comment: Figures 6, 8 and 11 need to be clearly seen.
Response: We thank the reviewer for the reminder. Indeed, there were clarity issues with these images. We have replaced these three images with higher-resolution versions.
Special thanks to you for your good comments.

Reviewer 3 Report
Coatings-2587350
1. The abstract should start with the most significant subject of the paper. The reader should understand the main topic of the paper from the first sentence. In addition, you need to focus more on quantitative information, not qualitative ones.
2. The authors should further highlight the novelty and significance of their work in the introduction section. Moreover, this section is not cohesive. Indeed, this section is intended to "convey the core findings of the paper," i.e., reflect the best novelty of this review paper in a concise form. The authors shall show the work's best novelty, such as how your research. advances the state-of-the-art of the topic/area and /or how much better is your work compared with peer researchers on the same or similar topics. At the end of this section, the main objective of this study must be mentioned.
3. Did you measure the bonding strength between coating and substrate?
4. More information regarding Ni-W alloy coating with heat treatment at 400℃, 600℃, and 900℃ must be given. You have to clarify the reasons for the wide range of heat treatment at 400℃, 600℃, and 900℃? You have to add a reference for that or explain how you got these heat treatment regimes (400℃, 600℃, and 900℃) as optimal ones.
5. Fig. 3 requires scale bar and scale value. In addition, the error band should be presented for Fig. 2 if the test has been conducted more than once. Similarly, Fig. 6 requires the error band if the test has been conducted more than once.
6. More information regarding Ni-W alloy coating at different current densities must be given. You have to clarify the reasons for wide range of current densities at 1, 2, 3, 4, and 5 A/dm2 ? You have to add a reference for that or explain how you got these current densities (1, 2, 3, 4, and 5 A/dm2) as optimal ones.
7. The coating thickness and uniformity of the coating has the most significant effect on the corrosion behavior of the substrate. In this respect, did you measure the coating thickness at different current densities?
8. The quality of Fig. 10 should be improved.
9. For the benefit of the readers I suggest that the author present the corrosion mechanism of Ni-W alloy coating at different current densities. More in-depth discussion of related previous works in this regard is required.
10. A reference about surface nickel-based materials may be useful for this article: Materials 2021, 14 (16), 4600. In addition, surprisingly small references to the Coatings in the literature despite the large relevant literature there. This should be improved. There are several important papers in recent literature.
Author Response
Dear Editors and Reviewers:
Thank you for your letter and for the reviewers’ comments concerning our manuscript entitled “The effect of heat treatment on phase and properties of high tungsten Ni-W alloy coating” (coatings-2587350). Those comments are all valuable and very helpful for revising and improving our paper, as well as the important guiding significance to our research. We have studied comments carefully and have made correction which we hope to meet with approval. Revised portions are marked in red in the paper. The main corrections in the paper and the responds to the reviewer’s comments are as flowing:
Responds to the reviewer’s comments:
Reviewer #1:
1.Comment: The abstract should start with the most significant subject of the paper. The reader should understand the main topic of the paper from the first sentence. In addition, you need to focus more on quantitative information, not qualitative ones.
Response: We appreciate the reviewer's suggestion. We have revised the abstract accordingly to make the topic clearer and have incorporated some quantitative conclusions.
- Comment: The authors should further highlight the novelty and significance of their work in the introduction section. Moreover, this section is not cohesive. Indeed, this section is intended to "convey the core findings of the paper," i.e., reflect the best novelty of this review paper in a concise form. The authors shall show the work's best novelty, such as how your research. advances the state-of-the-art of the topic/area and /or how much better is your work compared with peer researchers on the same or similar topics. At the end of this section, the main objective of this study must be mentioned.
Response: Based on the reviewer's suggestions, we have revised the introduction section to make the purpose of the article clearer and more cohesive.
- Comment: Did you measure the bonding strength between coating and substrate?
Response: We conducted a scratch test using a Rockwell indenter to measure the adhesion strength between the coating and the substrate. We found that whether in the as-plated state or after heat treatment, the adhesion remained strong with virtually no detachment of the coating. This could be attributed to the relatively thin coating and low internal stress we applied.
- Comment: More information regarding Ni-W alloy coating with heat treatment at 400℃, 600℃, and 900℃ must be given. You have to clarify the reasons for the wide range of heat treatment at 400℃, 600℃, and 900℃? You have to add a reference for that or explain how you got these heat treatment regimes (400℃, 600℃, and 900℃) as optimal ones.
Response: We appreciate the reviewer's pointed question. We chose 400°C, 600°C, and 900°C as the heat treatment temperatures based on several considerations. Firstly, these temperatures cover a range that allows us to study the performance variation of the coating at different temperature levels. Secondly, 400°C is a commonly used heat treatment temperature where certain phase changes or precipitation reactions are often observed. On the other hand, 600°C and 900°C represent higher temperatures that might lead to more significant microstructural changes and phase transformations.
Additionally, these temperature selections align with similar alloy systems' heat treatment temperatures in previous research and literature, enabling better comparison and analysis with existing studies. It's worth mentioning that we overlooked citing relevant references in our paper, and we apologize for this oversight. We have now included the appropriate references in the manuscript. Taking all these factors into account, we believe that selecting 400°C, 600°C, and 900°C as the heat treatment temperatures is justified and contributes to a deeper understanding of the performance change mechanisms in Ni-W alloy coatings.
- Comment: Fig. 3 requires scale bar and scale value. In addition, the error band should be presented for Fig. 2 if the test has been conducted more than once. Similarly, Fig. 6 requires the error band if the test has been conducted more than once.
Response: We appreciate the reviewer's reminder. We have added scale bars and scale values to Figure 3. The data in Figure 2 originally included error bars, but due to small variances, they might not have been very prominent. For Figure 6, we calculated the arithmetic average of three lines. Since there are multiple data points and lines, and to ensure a clear and aesthetically pleasing representation of the trend, we chose not to include error bars.
- Comment: More information regarding Ni-W alloy coating at different current densities must be given. You have to clarify the reasons for wide range of current densities at 1, 2, 3, 4, and 5 A/dm2 ? You have to add a reference for that or explain how you got these current densities (1, 2, 3, 4, and 5 A/dm2) as optimal ones.
Response: We selected these current densities (1, 2, 3, 4, and 5 A/dm²) based on multiple considerations. Firstly, we aimed to cover a certain range to investigate the impact of different current densities on the performance of Ni-W alloy coatings. Secondly, these current densities have been widely utilized in previous research, allowing for comparison and analysis with existing literature. Moreover, these values align with the common range used in industrial practices, facilitating the practical application of research findings.
Furthermore, we conducted preliminary experiments and simulation calculations to ensure that these chosen current densities would achieve the desired coating performance within appropriate ranges. Taking all these factors into account, we determined these current densities as the scope of our study to effectively explore the mechanism of current density influence on the performance of Ni-W alloy coatings.
- Comment: The coating thickness and uniformity of the coating has the most significant effect on the corrosion behavior of the substrate. In this respect, did you measure the coating thickness at different current densities?
Response: We considered the question raised by the reviewer at the initial stage of experimentation. Therefore, we measured the thickness of the coating at various current densities, as indicated below. We then selected different electroplating times at different current densities to achieve a consistent coating thickness(100mg). Due to constraints on the length of the paper, we regretfully omitted this part of the content.
- Comment: The quality of Fig. 10 should be improved.
Response: We have replaced the images with higher resolution versions and uploaded the source files of the images. Thank you for bringing this to our attention.
- Comment: For the benefit of the readers I suggest that the author present the corrosion mechanism of Ni-W alloy coating at different current densities. More in-depth discussion of related previous works in this regard is required.
Response: Thank you for the reviewer's suggestion. We have incorporated a discussion on the corrosion mechanisms under different current densities in the corrosion section of the paper.
- Comment: A reference about surface nickel-based materials may be useful for this article: Materials 2021, 14 (16), 4600. In addition, surprisingly small references to the Coatings in the literature despite the large relevant literature there. This should be improved. There are several important papers in recent literature.
Response: Thank you for the reviewer's reminder. We carefully read the article "Phase Formation during Heating of Amorphous Nickel-Based BNi-3 for Joining of Dissimilar Cobalt-Based Superalloys" and found that they studied the phase transformation and melting range of intermetallic BNi-3 during heating using differential scanning calorimetry (DSC), revealing three crystallization stages during the heating process. The presence of three exothermic peaks indicates solid-state crystallization. Their experimental results and analytical methods complement the discussion in our DSC section. In accordance with the reviewer's suggestion, we have added references to relevant coatings literature.
Special thanks to you for your good comments.

Reviewer 4 Report
Manuscript reports on systematic study of electrodeposition of NiW alloys. I general manuscript presents new and interesting results, nevertheless it requires serious rework to present the result clearly for readers.
1. Lines 74-75: ‘As deposited sample and coating powder samples were 74 placed in a vacuum tube furnace and protected by Ar gas’. Metalic Ni is very unstable to oxidation upon heating. Please, specify the residual oxygen content in the protective Ar gas and confirm that it is lower than the equilibrium pO2 for Ni/NiO.
2. Table 1. The composition of electrolyte is rather complex. Please, specify the origin and purity of all components. Names of organic components are necessary for clarity. In case the composition of electrolyte was taken from already published works the appropriate references are necessary.
3. Section 2, lines 80-106. Authors are strongly requested to present a detailed description of all instruments. The title, manufacturer and measurement conditions are mandatory information. For instance, for XRD analysis, the radiation wavelength, measurement geometry, beam configuration and presence of filters/monochromators should be specified.
4. Figure 1. The figure caption does not contain the legend for a-ad, a-400, … etc. notations. Please, add the legend or modify the figure to make it clearer.
5. Authors compare the elemental composition of samples prepared at different current densities (Figure 2) based on EDX analysis. However it is not clear if the coatings have identical thickness. It is well known for thin film analysis by EDX that thickness significantly affect the ‘measured’ composition.
6. Figure 3. The figure caption does not correspond to figure ‘heat treatment at (a)400℃, (a)600℃, and (a)900℃, showing’, while figure shows panels a-1, …, d-3. I guess that a-1 shows Ni mapping in as-obtained sample but it is not obvious. It is not cleat what current density was used for the presented samples.
7. In general, authors show rework figures to make them easy to read. Namely, In figure 5b the peak from Ni4W is virtually invisible due to overlapping with dash line. Figure 10 has (in PDF version) has dark gray background which complicate reading.
8. Table 2 contains the data with significantly overestimated precision.
I did not found serious issues with English. Some phrases may be improved to be clearer.
Author Response
Dear Editors and Reviewers:
Thank you for your letter and for the reviewers’ comments concerning our manuscript entitled “The effect of heat treatment on phase and properties of high tungsten Ni-W alloy coating” (coatings-2587350). Those comments are all valuable and very helpful for revising and improving our paper, as well as the important guiding significance to our research. We have studied comments carefully and have made correction which we hope to meet with approval. Revised portions are marked in red in the paper. The main corrections in the paper and the responds to the reviewer’s comments are as flowing:
Responds to the reviewer’s comments:
Reviewer #1:
1.Comment:Lines 74-75: ‘As deposited sample and coating powder samples were 74 placed in a vacuum tube furnace and protected by Ar gas’. Metalic Ni is very unstable to oxidation upon heating. Please, specify the residual oxygen content in the protective Ar gas and confirm that it is lower than the equilibrium pO2 for Ni/NiO.
Response: Thank you for the reviewer's suggestion. The equilibrium oxygen partial pressure (pO2) for nickel/nickel oxide depends on temperature. However, in the approximate range of 800°C to 1000°C, the equilibrium pO2 between nickel and nickel oxide is typically within the range of 10^-15 to 10^-18 atm. The argon gas we purchased is expected to be ultra-pure, but due to equipment limitations, we cannot determine its oxygen partial pressure. During the heat treatment process, even if there is a small amount of oxidation, it would occur on the surface of the coating and is not expected to have a significant impact.
- Comment: Table 1. The composition of electrolyte is rather complex. Please, specify the origin and purity of all components. Names of organic components are necessary for clarity. In case the composition of electrolyte was taken from already published works the appropriate references are necessary.
Response: Thank you for pointing out this issue. We have added the names of the organic components in Table 1. Additionally, the composition of this electrolyte is commonly used in the industrial application of Ni-W alloy, and it is considered a widely recognized formulation.
- Comment: Section 2, lines 80-106. Authors are strongly requested to present a detailed description of all instruments. The title, manufacturer and measurement conditions are mandatory information. For instance, for XRD analysis, the radiation wavelength, measurement geometry, beam configuration and presence of filters/monochromators should be specified.
Response: Thank you for the reviewer's suggestion. We have added descriptions of the model and manufacturer for all instruments in the section 2, and we have included the testing conditions for XRD analysis as well.
- Comment: Figure 1. The figure caption does not contain the legend for a-ad, a-400, … etc. notations. Please, add the legend or modify the figure to make it clearer.
Response: We apologize for such oversight, and we have made modifications in the figure legend to enhance clarity and understanding.
- Comment: Authors compare the elemental composition of samples prepared at different current densities (Figure 2) based on EDX analysis. However it is not clear if the coatings have identical thickness. It is well known for thin film analysis by EDX that thickness significantly affect the ‘measured’ composition.
Response: We considered the question raised by the reviewer at the initial stage of experimentation. Therefore, we measured the thickness of the coating at various current densities, as indicated below. We then selected different electroplating times at different current densities to achieve a consistent coating thickness(100mg). Due to constraints on the length of the paper, we regretfully omitted this part of the content.
6.Comment: Figure 3. The figure caption does not correspond to figure ‘heat treatment at (a)400℃, (a)600℃, and (a)900℃, showing’, while figure shows panels a-1, …, d-3. I guess that a-1 shows Ni mapping in as-obtained sample but it is not obvious. It is not cleat what current density was used for the presented samples.
Response: We apologize for such oversight, and we have made modifications in the figure legend to enhance clarity and understanding. Thank you for the reviewer's reminder. We have made modifications to these two figures to enhance their clarity.
7.Comment: In general, authors show rework figures to make them easy to read. Namely, In figure 5b the peak from Ni4W is virtually invisible due to overlapping with dash line. Figure 10 has (in PDF version) has dark gray background which complicate reading.
Response: Thank you for the reviewer's reminder. We have made modifications to these two figures to enhance their clarity.
8.Comment: Table 2 contains the data with significantly overestimated precision.
Response: Indeed, as pointed out by the reviewer, there was an issue with excessive precision. We have now adjusted the data in Table 2 to five decimal places.
Special thanks to you for your good comments.

Round 2
Reviewer 4 Report
Authors have addressed some reviewer’s comments, nevertheless the manuscript still requires rework in several points:
1. Line 5, ‘Firstname Lastname 1, Firstname Lastname 2 and Firstname Lastname ’Authors’ names should be presented instead.
2. Revised Table 1 contains wrong data: ‘Na3C6H5O7·2H2O (Saccharin)’, while the Saccharin is C7H5NO3S.
3. Figure 5 remains unclear in revised manuscript. Please, make all panels to be identical size and scale. Make the lines thick enough for well reading.
4. Table 2 still contains data with overestimated precision. Taking into account the data quality one could hardly expect the precision for RTC better than 0.001.
Author Response
Dear Editors and Reviewers:
Thank you for your letter and for the reviewers’ comments concerning our manuscript entitled “The effect of heat treatment on phase and properties of high tungsten Ni-W alloy coating” (coatings-2587350). Those comments are all valuable and very helpful for revising and improving our paper, as well as the important guiding significance to our research. We have studied comments carefully and have made correction which we hope to meet with approval. Revised portions are marked in red in the paper. The main corrections in the paper and the responds to the reviewer’s comments are as flowing:
Responds to the reviewer’s comments:
Reviewer #1:
1.Comment: Line 5, ‘Firstname Lastname 1, Firstname Lastname 2 and Firstname Lastname ’Authors’ names should be presented instead.
Response: Thank you to the reviewers for the corrections; we have revised the author's first and last names accordingly.
- Comment: Revised Table 1 contains wrong data: ‘Na3C6H5O7·2H2O (Saccharin)’, while the Saccharin is C7H5NO3S.
Response: We apologize for making such a mistake. The chemical name should be Trisodium citrate, and it has been corrected in the table.
- Comment: Figure 5 remains unclear in revised manuscript. Please, make all panels to be identical size and scale. Make the lines thick enough for well reading.
Response: Thank you to the reviewers for the suggestion; we have redrawn Figure 5 in accordance with the reviewers' recommendations.
- Comment: Table 2 still contains data with overestimated precision. Taking into account the data quality one could hardly expect the precision for RTC better than 0.001.
Response: Thank you to the reviewers for the correction; we have revised the data in the table to retain three decimal places.
Special thanks to you for your good comments.
